# KSNP: a fast de Bruijn graph-based haplotyping tool approaching data-in time cost

Qian Zhou [1,6], Fahu Ji [2,6], Dongxiao Lin[3,6], Xianming Liu [1,2], Zexuan Zhu [3,4] ✉ & Jue Ruan [5] ✉

Long reads that cover more variants per read raise opportunities for accurate haplotype construction, whereas the genotype errors of single nucleotide polymorphisms pose great computational challenges for haplotyping tools. Here we introduce KSNP, an efficient haplotype construction tool based on the de Bruijn graph (DBG). KSNP leverages the ability of DBG in handling high-throughput erroneous reads to tackle the challenges. Compared to other notable tools in this field, KSNP achieves at least 5-fold speedup while producing comparable haplotype results. The time required for assembling human haplotypes is reduced to nearly the data-in time.

Haplotyping is the process of distinguishing the alleles that are inherited together on a chromosome from a parent in a diploid or polyploid genome. Haplotyping is not only crucial for interpreting the genetic mechanism underlying biological phenotypes but also a non-negligible step in heterozygous genome assembly and variant detection[1–3]. As a traditional bioinformatics analysis, constructing haplotypes faces the great challenge of dealing with error-prone single nucleotide polymorphism (SNP) genotypes present in reads due to sequencing and mapping errors. Haplotyping has been formulated as several computational problems by bioinformaticians, as reviewed in ref. 4. Among these formulations, minimum error correction (MEC) is probably the most popular one and has been implemented by many successful haplotyping methods[5]. The existing MEC-based methods attempt to find a genotype combination to represent a potential haplotype that maximizes consistency with the observed reads by flipping as few SNP genotypes as possible.

The third-generation sequencing technologies produce long reads spanning tens to hundreds of kilobases (kb), providing opportunities for haplotype construction. With long reads, a single read can cover more variant sites, making it possible to generate more accurate genome-scale haplotypes. MEC-based heuristics, for example, Marginphase[6], WhatsHap[7,8], and HapCUT2[9,10] have shown promise in

accurately reconstructing haplotypes using long reads at a genome-wide scale. However, with the growth of read length and data volume, the computational burden of these model-based methods increases dramatically. For example, the time complexities of the WhatsHap and HapCUT2 are $O(N2^d)$ ($d \leq 15$) and $O(M\log(N)+NdV^2)$, respectively, where $N$ is the total number of variants, $d$ is the maximum coverage per variant, and $V$ is the maximum number of variants per read. When $d$ or $V$ reaches tens or over a hundred, which is very common on Mb-level ultra-long reads, the time complexity increases considerably. In the era of decreasing sequencing cost and the rapid development of precision medicine, a large number of human genomes are being sequenced, still requiring more computationally efficient haplotyping approaches.

In this study, we use strings of SNPs as "pseudo-read" and employ efficient graph-based assembly algorithms, which have been well-developed in theory and practice, to solve the haplotype construction problem. We present a de Bruijn graph (DBG)-based tool, called KSNP, for haplotype construction and demonstrate that it can generate human haplotypes from aligned PacBio Continuous Long Reads (CLR), PacBio High Fidelity (HiFi) reads or Oxford Nanopore Technologies (ONT) reads in only ~30 min. An analysis of the time cost for each processing step reveals that 90% of the time is spent on the inevitable reading and decompression of data from BAM (Binary Alignment/Map) files.

[1]PengCheng Laboratory, Shenzhen, China. [2]School of Computer Science and Technology, Harbin Institute of Technology, Harbin, China. [3]College of Computer Science and Software Engineering, Shenzhen University, Shenzhen, China. [4]National Engineering Laboratory for Big Data System Computing Technology, Shenzhen University, Shenzhen, China. [5]Shenzhen Branch, Guangdong Laboratory for Lingnan Modern Agriculture, Genome Analysis Laboratory of the Ministry of Agriculture and Rural Affairs, Agricultural Genomics Institute at Shenzhen, Chinese Academy of Agricultural Sciences, Shenzhen, China. [6]These authors contributed equally: Qian Zhou, Fahu Ji, Dongxiao Lin. ✉e-mail: zhuzx@szu.edu.cn; ruanjue@caas.cn

## Results

### Overview of KSNP

Following the principle of constructing DBG[11], KSNP uses a sliding approach to extract $k$ consecutive SNPs from the reads to form the $k$-mers, which are characterized by the genomic positions and genotypes of the SNPs (Fig. 1). Benefiting from the uniform sequencing coverage and the length of the long reads, the constructed DBG exhibits strong connectivity, with connected components that typically span long genomic regions. However, erroneous k-mers containing incorrect SNP genotypes arising from the sequencing or mapping errors make almost all vertices associated with competing edges on the graph and pose the biggest challenge to obtain unambiguous haplotypes. To simplify the graph efficiently, KSNP applies a four-step greedy pruning heuristic (Fig. 2), i.e., (1) fast pruning of the initial graph by cutting off the competing edges with low weights; (2) removing short tips by traversing the graph in both forward and backward directions; (3) identifying the optimal or near-optimal path in bubble regions through the comparison between the MEC scores of the competing paths; and (4) processing the remain long branches by completing them into

bubbles. After linear traversing, the graph can be quickly simplified to contain only unambiguous paths, from which the haplotypes are plainly constructed. More implementation details of KSNP are provided in "Methods" and the pseudocode of the graph pruning is outlined in Supplementary Note 1.

### Performance of KSNP

The $k$-mer size determines the connectivity and the edge depth of the haplotype graph. Currently, KSNP supports k-values ranging from 2 to 5. A larger $k$-mer size increases the reliability of the edges while decreasing the edge depths and the connectivity of the graph. It is conceivable that a larger $k$ value results in higher haplotype accuracy, whereas a smaller $k$ value results in a higher recall rate and longer haplotypes. This was demonstrated in our experiments using eight CLR and ONT datasets. Using 5-mer on HG001, HG002, and HG005 CLR datasets, which have averagely 6.6, 9.6, and 12.6 heterozygous SNPs per read, respectively, resulted in a 2–3% lower recall rate than that of using 2-mer (Supplementary Tables 1 and 3). However, for HG01109 and *A.thaliana* F1 CLR reads, which averagely contain 23.3

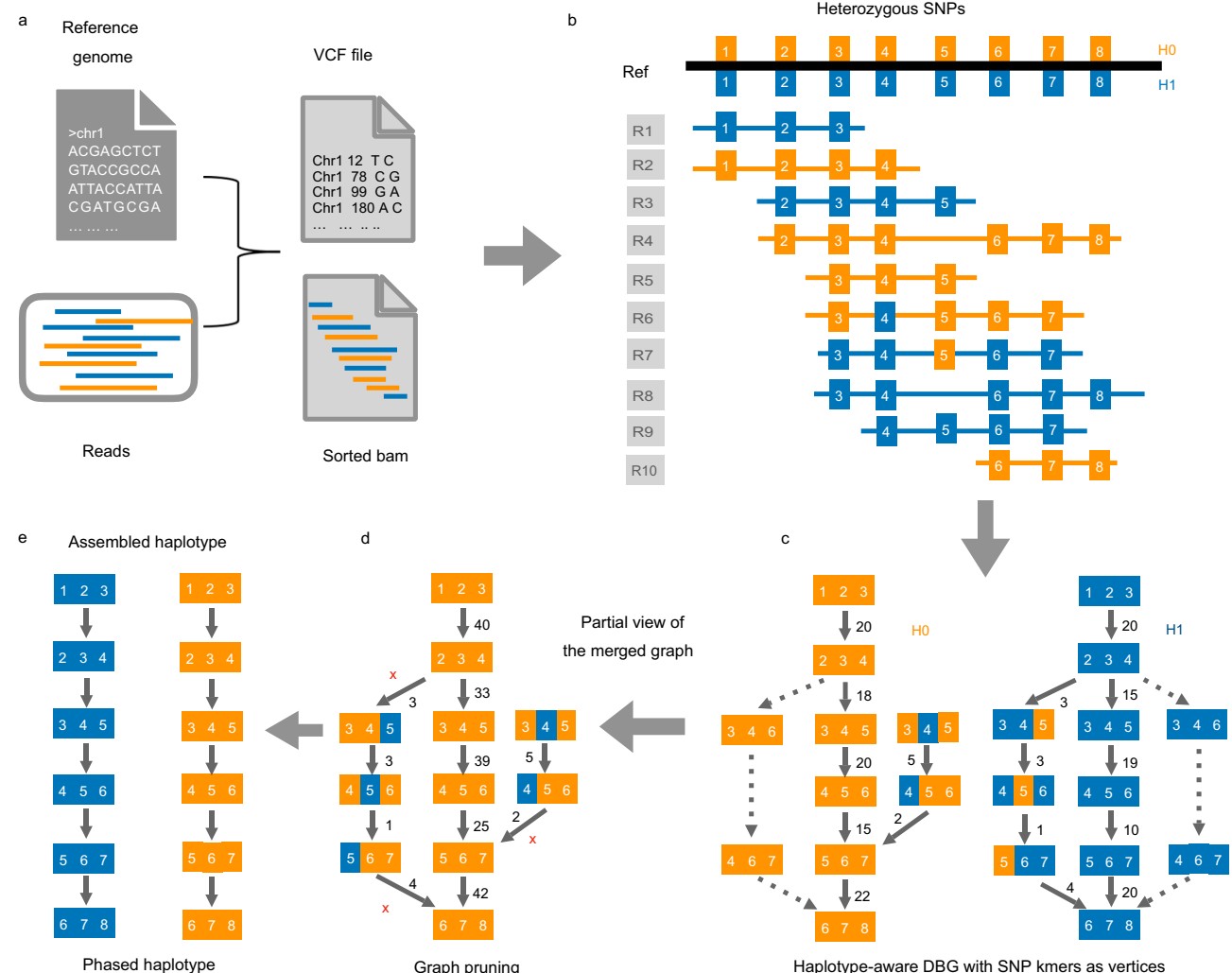

**Fig. 1 | Overview of KSNP algorithm.** KSNP is currently implemented for diploid genome. **a** The input files of KSNP. **b** Heterozygous SNPs (blue and orange rectangles) detected on reads. **c** The featured k-mers (k-SNPs) extracted from SNP strings on long reads (R1, R2,...), e.g., $k = 3$, are used to construct the DBG. The transitive edges (dotted arrows) caused by the deletion errors in reads are excluded in the graph construction. A number marked on an edge represents the corresponding edge depth, equivalent to the number of reads supporting the edge.

**d** Considering the inherent complementarity of the two haplotypes in diploid genome, the edges in two subgraphs are merged. The merged graph is self-symmetric and retains all the information of the two haplotypes H0 and H1. Bubbles and small tips caused by the genotype errors in reads are heuristically pruned during graph traversal. **e** After graph pruning, one haplotype is assembled directly from the graph, and the other is obtained by complementation.

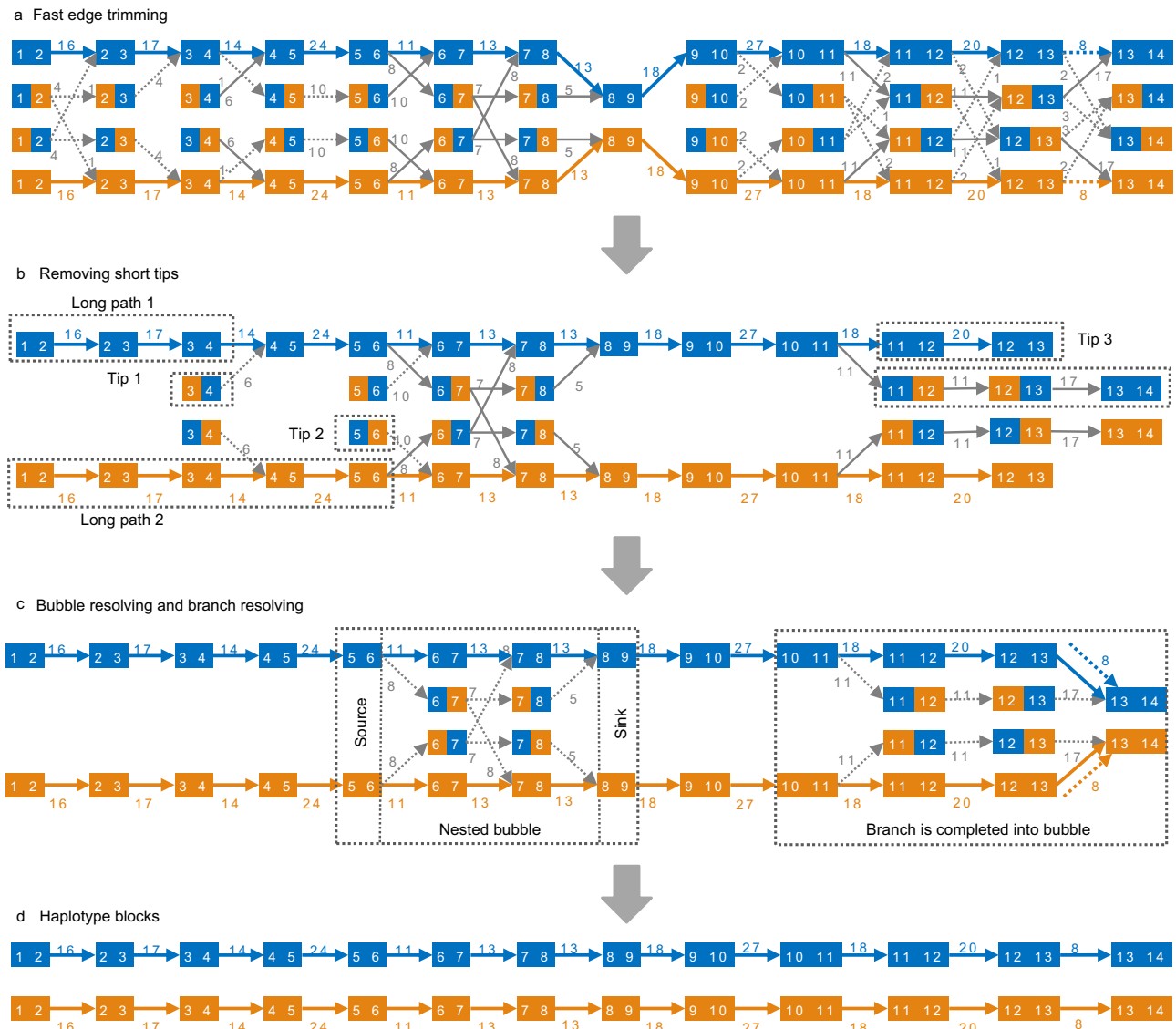

**Fig. 2 | A schematic diagram of graph pruning in KSNP ($k = 2$).** The blue and orange nodes represent 2-mers from haplotypes H0 and H1, respectively, where the two numbers in a node indicate the corresponding position indices of the two consecutive SNPs. The solid and dashed arrows represent edges that are retained and removed in each processing step, respectively. The numbers on the arrows denote the depth of the edges. This graph is horizontally symmetrical, indicating the two complementary haplotypes in a diploid genome. **a** The competitive edges (dashed arrows) at the same SNP position are removed since they have very shallow depth in comparison with the dominate path. **b** The short tip is deleted if the length of the long path exceeds three times that of the short path. For example, Tip1 and Tip2 are deleted given Long path 1 and Long path 2, respectively. **c** For a bubble, KSNP attempts to find an optimal path from the source node to the sink node under the supervision of MEC scores. For example, KSNP calculates the MEC scores of the three possible haplotypes in the illustrated nested bubble using the reads involved in the bubble (not displayed), and the nodes and edges that are not located on the best path are removed. For a long branch like the one from node [10,11] to node [13,14], by restoring a previously removed edge between node [12,13] and node [13,14], the branch can be completed into a bubble and then resolved as a bubble. **d** The haplotype blocks are generated by walking out the paths in the pruned graph.

and 36.0 heterozygous SNPs per read, respectively, using 5-mer only caused a 0.2% and 0.01% drop, respectively, while also reducing hamming errors. On ONT reads that have longer read length but relatively higher error rate than CLR reads, we tested KSNP using SNPs identified by two variant callers, i.e., Longshot[12] and PEPPER-Margin-DeepVariant (PMD)[13]. Compared with Longshot, PMD reported fewer but more accurate SNPs. For SNPs identified by these two variant callers, using 5-mer significantly reduced the hamming errors at the cost of less than 0.3% loss of recall rate. Except for the *A. thaliana* CLR data, a larger *k* value consistently decreased the obtained haplotype N50 length during the experiments. If better haplotype continuity is a priority, a smaller *k* value is more desirable (Supplementary Tables 4 and 5).

The time complexity of KSNP can be expressed as $O(Nd(logN + V))$ (details in "Methods"). The running time of KSNP is linear, with the maximum read depth and the maximum variants per read, giving it a theoretical advantage in handling high-throughput long reads. We evaluated KSNP along with four widely used haplotyping tools, namely Longshot[12] (v0.4.1), Whatshap[7,8] (v1.7), HapCUT2[9,10] (v1.3.1), and Margin (v2.3.1) on human and heterozygous *A. thaliana* genomes. Overall, the five tools generated comparable results on the eight datasets examined. In terms of accuracy, Margin and Longshot performed the best, whereas considering haplotype length and recall, WhatsHap, Hap-CUT2, and KSNP are the better methods. In terms of computational resource consumption, KSNP consumed the lowest peak memory and CPU time, which amount to only 1.2%-19.4% of other tools

**Table 1 | Performance of KSNP and the four state-of-the-art haplotyping tools on PacBio CLR and HiFi datasets**

| Dataset[22,25] | Tool[a] | SE (%) | HE (%) | Hap N50 (kb) | Recall (%) | CPU time (s) | Wall time (s) | RAM (MB) |
|---|---|---|---|---|---|---|---|---|
| HG001 CLR 50× | Longshot | 0.63 | 1.14 | 239 | 91.56 | 13,485 | 14,186 | 2970 |
| | WhatsHap | 0.67 | 1.57 | 241 | 91.61 | 20,714 | 22,050 | 1843 |
| | HapCUT2 | 0.67 | 1.84 | 251 | 91.66 | 6100 | 6478 | 969 |
| | Margin | 0.63 | 0.89 | 171 | 90.30 | 146,551 | 19,283 | 2867 |
| | KSNP[a] | 0.68 | 1.82 | 246 | 91.59 | 1881 | 2004 | 510 |
| HG002 CLR 50× | Longshot | 1.22 | 1.53 | 315 | 89.86 | 13,537 | 14,267 | 2662 |
| | WhatsHap | 1.25 | 2.03 | 317 | 89.91 | 19,141 | 19,963 | 1536 |
| | HapCUT2 | 1.25 | 2.25 | 324 | 89.95 | 6042 | 6453 | 802 |
| | Margin | 1.21 | 1.38 | 244 | 89.17 | 128,479 | 16,471 | 2458 |
| | KSNP | 1.25 | 2.12 | 321 | 89.91 | 1607 | 1740 | 476 |
| HG002 HiFi 50× | Longshot | 1.26 | 1.21 | 416 | 93.67 | 11,285 | 11,982 | 2458 |
| | WhatsHap | 1.27 | 1.48 | 417 | 93.68 | 13,254 | 13,880 | 1229 |
| | HapCUT2 | 1.27 | 1.41 | 417 | 93.68 | 5125 | 5483 | 685 |
| | Margin | 1.26 | 1.17 | 370 | 93.42 | 112,980 | 15,064 | 2253 |
| | KSNP | 1.27 | 1.46 | 422 | 93.68 | 1106 | 1270 | 462 |
| HG005 CLR 50× | Longshot | 1.45 | 2.77 | 490 | 92.75 | 17,968 | 18,865 | 2765 |
| | WhatsHap | 1.49 | 4.01 | 507 | 92.80 | 22,968 | 24,093 | 1434 |
| | HapCUT2 | 1.49 | 4.43 | 528 | 92.84 | 8202 | 8659 | 946 |
| | Margin | 1.44 | 2.40 | 315 | 91.23 | 145,455 | 18,648 | 2253 |
| | KSNP | 1.49 | 3.98 | 514 | 92.80 | 2088 | 2268 | 513 |
| HG01109 CLR 40× | Longshot | 0.04 | 4.92 | 5080 | 89.61 | 20,794 | 21,930 | 2662 |
| | WhatsHap | 0.06 | 5.42 | 5323 | 89.66 | 22,919 | 23,917 | 1331 |
| | HapCUT2 | 0.07 | 6.11 | 5717 | 89.67 | 9615 | 10,184 | 849 |
| | Margin | 0.03 | 2.46 | 3804 | 89.47 | 125,506 | 16,513 | 2355 |
| | KSNP | 0.07 | 5.94 | 5333 | 89.66 | 1304 | 1449 | 512 |
| *A.thaliana* CLR 45× | Longshot | 0.01 | 2.59 | 4001 | 85.97 | 3570 | 3586 | 911 |
| | WhatsHap | 0.01 | 2.12 | 4001 | 85.98 | 2610 | 2655 | 615 |
| | HapCUT2 | 0.01 | 2.43 | 4001 | 85.99 | 1753 | 1759 | 392 |
| | Margin | 0.01 | 2.07 | 1228 | 85.37 | 9619 | 1202 | 2150 |
| | KSNP | 0.01 | 2.56 | 4001 | 85.98 | 100 | 100 | 152 |

*SE* switch error rate, *HE* hamming error rate, *Wall time* wall clock time, *RAM* peak RAM.
[a]Longshot, WhatsHap, HapCUT2, and KSNP were performed with one thread in the experiments, while Margin utilized eight threads. The *k* value in KSNP was set to two by default.

(Tables 1 and 2). To assess the end-to-end wall clock time, we employed one thread for single-threaded tools i.e., Longshot, WhatsHap, Hap-CUT2, and KSNP, and eight threads for Margin, where KSNP exhibited a speed advantage of 5.0–11.0 times. Upon examining the running steps of KSNP, we discovered that reading and decompressing of BAM and VCF files accounted for ~83.5% of the end-to-end wall clock time, whereas constructing and resolving graph accounted for only 5.7%. For example, on 50× HG001 ONT data, the read-in and decompression time of the input was 33 min, yet the graph processing and haplotype construction time was only 2 min (Supplementary Table 6). The observation suggests that the speedup of KSNP on haplotype construction had been pushed to the limit. When the sequencing depth is relatively low, such as ~20×, KSNP can still maintain good performance (Supplementary Tables 7 and 8).

To investigate the impact of read types on haplotype construction, we conducted experiments on Illumina, CLR, ONT, and HiFi datasets of HG002 genome (Supplementary Table 9) using three fast haplotyping tools, i.e., Longshot, HapCUT2 and KSNP. The use of HiFi reads led to a higher recall rate when phasing SNPs identified from the long reads themselves (HiFi vs CLR vs ONT: 93.7% vs 89.9% vs 89.8% in average). However, this advantage disappeared when the long reads were used to phase SNPs called from Illumina reads (HiFi vs CLR vs ONT: 94.0% vs 93.5% vs 95.2% in average). Compared to CLR and HiFi reads, ONT reads were capable of generating 30 times longer

haplotypes in terms of N50 length, with a similar switch error rate and a 4 times higher hamming error rate. Therefore, using accurate reads or longer reads for constructing haplotypes should be decided based on the user's requirements for haplotype accuracy and length.

## Discussion

Since the PacBio reads were used to phase the variants on the human genome in 2015 *in* ref. 14, the read length and sequencing throughput of the third-generation reads have increased by dozens of times. As the amount of data increases, users typically use more threads or develop hardware acceleration solutions to shorten the analysis time. More efficient algorithms are also desirable to reduce the computational costs associated with the pipeline (Supplementary Fig. 1). In this study, KSNP provides an ultra-fast haplotyping algorithm that can save both the time and computational costs of phasing analysis. The success of KSNP is attributed to the abilities of DBG in handling sequencing reads, i.e., (1) DBG can efficiently capture the SNP-k-mer information present in large quantities of reads, even they contain errors, in a concise graph format; (2) the prune-search algorithm employed in DBG offers a low-complexity approach to rapidly determine the correct path; and (3) the multiple sources of evidence in DBG, including edge weight, path length, and MEC score can guarantee the reliability of the resulting haplotype.

The simplicity and computational efficiency of KSNP make it qualified for various haplotype-aware tasks, such as genome assembly,

**Table 2 | Performance of KSNP and the four state-of-the-art haplotyping tools on ONT datasets**

| Dataset | Tool[a] | SE (%) | HE (%) | Hap N50 (kb) | Recall (%) | CPU time (s) | Wall time (s) | RAM (MB) |
|---|---|---|---|---|---|---|---|---|
| HG001 ONT 50× | Longshot | 0.66 | 3.27 | 4576 | 91.64 | 21,249 | 22,540 | 2765 |
|  | WhatsHap | 0.67 | 5.14 | 4902 | 92.36 | 19,989 | 21,018 | 1434 |
|  | HapCUT2 | 0.67 | 4.64 | 5208 | 92.36 | 10,284 | 10,930 | 867 |
|  | Margin | 0.66 | 1.13 | 1504 | 91.97 | 151,309 | 19,650 | 2253 |
|  | KSNP | 0.67 | 5.37 | 4921 | 92.36 | 2091 | 2307 | 486 |
| HG002 ONT 50× | Longshot | 1.23 | 5.38 | 13,074 | 89.21 | 30,954 | 32,919 | 2662 |
|  | WhatsHap | 1.24 | 5.76 | 13,103 | 90.03 | 23,909 | 25,274 | 1229 |
|  | HapCUT2 | 1.24 | 6.08 | 13,103 | 90.03 | 15,645 | 16,652 | 842 |
|  | Margin | 1.23 | 3.65 | 7590 | 89.73 | 169,673 | 22,623 | 1741 |
|  | KSNP | 1.24 | 6.18 | 13,103 | 90.03 | 2265 | 2536 | 476 |
| HG005 ONT 50× | Longshot | 1.45 | 5.50 | 7400 | 89.50 | 32,509 | 34,514 | 2765 |
|  | WhatsHap | 1.46 | 7.75 | 8257 | 90.42 | 22,990 | 24,146 | 1229 |
|  | HapCUT2 | 1.46 | 6.46 | 8621 | 90.42 | 16,633 | 17,724 | 973 |
|  | Margin | 1.45 | 3.13 | 3757 | 90.11 | 192,759 | 24,712 | 1741 |
|  | KSNP | 1.46 | 6.72 | 8388 | 90.42 | 2558 | 2866 | 505 |

*SE* switch error rate, *HE* hamming error rate, *Wall time,* wall clock time, *RAM* peak RAM.
[a]Longshot, WhatsHap, HapCUT2 and KSNP were performed with one thread in the experiments, while Margin utilized eight threads. The *k* value in KSNP was set to two by default.

genome polishing, and structural variant calling. As reviewed in refs. 15,16, chromosome-scale haplotyping relies on not only long reads but sometimes long-range short reads, such as Hi-C data. Therefore, haplotyping software is more favorable to possess the ability to handle evidence from multiple sequencing technologies. The haplotype-aware graph constructed by KSNP is suitable for incorporating evidence from different data sources, providing more information for the graph pruning step. In future developments, we will focus on enhancing the functionality of KSNP to handle multiple types of data. For example, the Hi-C sequencing data and genetic data can also be used to weight the edges, providing more evidence in graph traversing and simplification. We also expect that KSNP can be extended to handle polyploid genomes, as graph is an ideal data structure for capturing the similarity and divergence among multiple haplotypes.

## Methods
### Graph processing in KSNP
**Read re-alignment.** The input files of KSNP include a sorted BAM[17] file containing the aligned reads and a VCF file containing the genotypes of heterozygous variants. To minimize the SNP genotype errors introduced by sequencing or mapping, local re-alignment is adopted in KSNP. Particularly, KSNP extracts a 31 bp sequence around a SNP (i.e., including 15 bases before and another 15 bases after the SNP) from the read, and aligns the extracted sequence to two target sequences. One is from the corresponding aligned window in the reference, and the other is generated by replacing the reference allele (center of the window) with the alternative SNP allele. The SNP allele with a smaller edit distance is taken as credible. Efficient Myer's bit-vector algorithm is used to calculate the alignment score where each column is represented by one 32-bit word[18].

**Graph construction.** The corrected SNP genotypes of reads are read as strings, from which the k-mers are derived with the genomic positions and genotypes as unique identifiers. Each featured k-mer initially forms a vertex on the graph. The identical featured *k*-mers are collapsed into a single vertex. A $(k+1)$-mer on a read introduces an edge between its prefix and suffix *k*-mer vertexes, and the number of the reads bridging the two connected vertices defines the edge depth. Considering the natural symmetry of two haplotypes in diploid genome, the complement of vertices and edges are generated at the same time to reduce the imbalance of sequencing depth. As such, the graph is internally symmetrical (Fig. 1).

**Graph pruning.** The genotype errors introduced in sequencing or mapping are encoded in ambiguous paths, which would be screened out in the traversal of the graph. KSNP sequentially performs the following steps to prune the graph (illustrated in Fig. 2):

(1) *Fast edge trimming:* The initial graph tends to contain a large fraction of erroneous edges with very low depth. Among the $2^{k+1}$ competing edges at the same SNP positions that represent $2^k$ different phasing solutions, the shallow-depth edges are removed in comparison with the maximum edge depth. In particular, if the maximum edge depth *M* is larger than a preset value *C*, a competing edge with a depth less than *M*/2 is cut off. Otherwise, an edge with a depth less than *M*/5 is removed. *C* is set to 15 by default in KSNP.

(2) *Tips removing:* Short branching paths that are unable to extend into long haplotypes blocks are also removed. More precisely, given two linear paths starting from the same vertex in the forward or backward direction, if their lengths are different by a factor of 3, the shorter one is removed. To process forwardly branching paths, KSNP starts from the last vertex in graph and walks the graph backwardly. In this way, KSNP can reduce the possibility of two inspected paths branching again and improve the efficiency of identifying short paths. Similarly, KSNP processes backwardly branching paths by taking the first vertex as the starting point and walking the graph in the forward direction.

(3) *Bubbles resolving:* After faster trimming, the major unresolved issues in the remaining graph are the bubble structures. Typically, a bubble begins at source node *s* with two outdegrees and ends at a sink node *t* with two indegrees. Between *s* and *t*, there are two disjoint paths which encodes two different phasing solutions in this genome region[19]. A supper bubble might contain nested bubbles and present more complexity of phasing. Although it is not strictly accurate, we refer to both of these structures as "bubbles" for the sake of convenience in the subsequent description of the processing steps. For each bubble, KSNP attempts to find an optimal path from the source node to the sink node with the minimum MEC score. If the number of paths in a bubble is fewer than 512, we can consider every possible solution and calculate its MEC score using the involved reads. The path with the minimum MEC score is the optimal path, and all other paths are removed. Nevertheless, such brute-force method is impractical for a complicated bubble that might contain the exponential number of paths due to its nesting structure. We adopt a heuristic

algorithm to solve the complex bubbles. We first choose the path with the maximum weight sum by dynamic programming (DP) in linear time, and then iteratively modify the path by replacing its edges with alternatives that could produce better MEC scores. We design a list of templates and follow them to switch on some alternative edges (Supplementary Note 2). Path updating stops until the MEC score is no longer improved or the number of iterations exceeds 512. In most cases, the calculation of MEC score converges after tens of iterations.

MEC score is calculated based on the reads involved in the bubble. However, in some cases, the read sets of multiple bubbles have intersections, meaning that a read is involved in more than one bubble. In such cases, it is impossible to calculate the MEC scores of individual bubbles separately, so the bubbles that have intersecting read sets are combined into to a larger bubble and solved as a whole.

(4) *Branches resolving:* After going through the previous steps, there might remain branching paths with relatively long length in the graph. These branching paths are very likely incomplete bubbles, with missing edges due to insufficient sequencing or previously trimming. By restoring essential previously removed edges on the graph, the branches could be connected to the main path, forming bubble structures, which can then be resolved as Step (3).

**Haplotype block generation.** After graph pruning, the primary haplotype blocks are generated by walking out the paths in the pruned graph. The spurious short blocks are filtered out if they are contained in a large block. To improve the haplotype continuity, blocks with overlapped SNPs are joined together if they are spanned by long reads. MEC score is used to determine which of the two complementary haplotypes to connect.

**Time complexity of KSNP**

In the *read re-alignment* of KSNP, the Myer's bit-vector algorithm is adopted to the calculation of DP matrix where each column can be solved in $O(1)$ time if the size of query sequence is smaller than the machine word size. Both the query and target sequence length are set to 31 bp. Because the scale of DP matrix is constant, the only factor in re-alignment complexity is the number of DP matrices. $Nd$ can represent the total number of reads, therefore, the time spent on re-alignment is $O(NdV)$, where $N$ is the total number of variants, $d$ represents the maximum coverage per variant, and $V$ indicates the maximum number of variants per read.

In *DBG construction*, the time spent is $O(Nd(\log N + V))$, where $LogN$ indicates the time of binary searching of the first allele on a read, and $V$ SNPs are taken as k-mers. In *fast trimming edges* and *removing tips*, the time complexity is linear to the number of edges. The graph contains at most $2^{k+1}(N-k)$ edges, in which $k$ is a fixed parameter less than 5. The time complexity of linear traversal on graph is $O(N)$. In *bubble resolving*, the reads involved are used to calculate MEC score for each candidate path in the bubbles. The number of examined paths in a bubble is limited to a constant threshold. In the worst-case scenario, where all reads are involved, the time complexity required for bubble resolving is $O(NdV)$. Overall, the time complexity of KSNP is $O(Nd(\log N + V))$.

**Evaluation metrics of haplotype construction**

To evaluate the performances of the haplotype assemblers, we used six criteria including switch error rate, hamming error rate, haplotype N50, recall rate, CPU time, and the peak RAM consumption. A switch error occurs when the phase between two adjacent SNPs in the assembled haplotype is discordant compared with the true haplotype. The switch error rate is calculated as the number of switch errors divided by the total number of phased SNPs minus one. The hamming error rate is the percentage of wrongly phased SNP sites in a haplotype against the true paternal or maternal haplotype. For example, given an

assembled haplotype "01000" and the true haplotype "00000", the switch error rate is 2/(5-1) and the hamming error is 1/5. The recall rate is calculated as the number of correctly phased SNPs divided by the number of all phased SNPs in ground truth dataset. The length of a haplotype block is the distance between its first and last phased SNPs. The haplotype N50 is the length $L$ of the haplotype block for which 50% of the total length of blocks are of length greater than $L$. The value is calculated by sorting the blocks from the largest to the smallest and accumulating them one by one until they reach 50% of the total length.

**Data processing in the experiments**

For HG001, HG002, and HG005 samples, high-confidence phased variants were extracted from Genome in a Bottle (GIAB)[20] files (Supplementary Table 10) as ground truth data. For HG01109, the Illumina reads of two parents were mapped to GRCh37 reference genome using BWA-MEM[21] (v0.7.17), and the parental SNPs were identified using bcftools mpileup pipeline. For *A.thaliana* F1 (Col-0 × Cvi-0) sample[22], because the TAIR10 reference genome was constructed based on Col-0 sample, the SNPs between Cvi-0 and the TAIR10 could be used as ground truth to evaluate the haplotype assembly of *A. thaliana* F1. The PacBio long read of Cvi-0 were aligned to the TAIR10 reference genome using BLASR[23] (v5.1) and the homozygous SNPs were identified using Longshot[12] (v0.4.1).

All downloaded datasets were randomly sampled to 45×–50× coverage of the genomes and aligned to the haploid reference genomes using BLASR[23] and minimap2[24] with default parameters. Besides, for CLR data, there was no significant difference between the final haplotype results based on aligner BLASR[23] and minimap2[24]. Minimap2 is recommended when running KSNP for the sake of speed (Supplementary Tables 2 and 3).

For CLR and HiFi reads, the heterozygous SNPs were identified using Longshot (v0.4.1, options --no_haps --max_cov 500). For ONT reads, besides Longshot, the PMD pipeline were also performed to identify SNPs (release r0.7, options --ont_r9_guppy5_sup). Only the heterozygous variants satisfying the following criteria were used in the phasing experiments: (1) with "PASS" tag in FILTER filed; (2) the DP4 depth is 0.5×–2× of the average sequencing depth; (3) with "0/1" tag in GT filed; and 4) the frequency of ALT allele is within 20–80%. We chose PMD as the SNP caller for ONT reads in the follow-up experiments (Supplementary Tables 4 and 5).

The experiments in this study were conducted on a high-performance computing cluster node with 24 cores. To accelerate the experimental processes, we split the input BAM and VCF files of the human genome into 22 parts by chromosomes and accordingly submitted 22 separate computational tasks. For single-threaded tools such as WhatsHap, HapCUT2, Longshot, and KSNP, all the 22 tasks were submitted simultaneously. As for Margin, we used eight threads within each task, and the 22 tasks were executed sequentially. The CPU time and end-to-end wall clock time for each task were recorded using the "/usr/bin/time" command.

**Reporting summary**

Further information on research design is available in the Nature Portfolio Reporting Summary linked to this article.

## Data availability

All the datasets used for evaluation were obtained from public databases. The accession numbers and data links are listed in Supplementary Table 10. The command lines used in this study are provided in Supplementary Note 3.

## Code availability

KSNP code is available at github: https://github.com/zhouqiansolab/KSNP. The version of KSNP used in this study can also be accessed through a permanent link https://doi.org/10.5281/zenodo.10863978.

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

## Acknowledgements

This study was supported by the National Key Research and Development Project Program of China (2019YFA0707003 and 2019YFE0109600), the Natural Science Foundation of China (31822029 and 61871272), and The Major Key Project of Pengcheng Laboratory (No. PCL2023AS7-1). The Peng Cheng Cloud-Brain Supercomputer provided the necessary computing resources for this study. We thank the Yongyao Li from the Agricultural Genomics Institute at Shenzhen for providing technical support for high-performance computing platforms.

## Author contributions

J.R. conceived the idea of KSNP and Q.Z. proofed the feasibility of the idea. J.R. and Z.Z. coordinated and supervised the project. Q.Z. and D.L. developed KSNP algorithm by Python, F.J. adopted KSNP in C + + and optimized the algorithms to improve the performance. Q.Z., D.L. and F.J. performed experiments and drafted the manuscript. Z.Z. and X.L. provided critical comments on algorithm evaluations and improved the manuscript.

## Competing interests

The authors declare no competing interests.
