## [Peer Review File · Nature Communications]

KSNP: a fast de Bruijn graph-based haplotyping tool
approaching data-in time costEditorial Note: This manuscript has been previously reviewed at another journal that is not operating a transparent peer review scheme. This document only contains reviewer comments and rebuttal letters for versions considered at *Nature Communications*. Mentions of prior referee reports have been redacted.

Reviewer #1 (Remarks to the Author):

[Redacted] I appreciate the inclusion of PacBio HiFi reads in the benchmark, as well as Supplementary Figure 1. I continue to believe that the proposed approach is novel with respect to existing methods for phasing. However, I am still not convinced if KSNP offers sufficient empirical advantage over the existing state-of-the-art.

Authors have presented the following argument in the Introduction section to motivate this work:

"As the amount of data increases, users typically use more threads or develop hardware acceleration solutions to improve the speed of read alignment and variant detection. However, The current phasing tools are still unable to fully utilise the parallel computing power of hardware., making it difficult to shorten the analysis time and gradually becoming an inefficient step in the pipeline"

The main results (Table 1 and Table 2) presented in the manuscript are based on sequential single-threaded runs of the existing phasing tools. However, the authors appear to ignore the fact that Margin supports efficient multi-threading. As a result, the performance evaluation has been done by using only a single thread. This evaluation is unfair because Margin would have used significantly less time with multiple threads. Please check:

<https://github.com/UCSC-nanopore-cgl/margin#resource-requirements>

Additionally, other tools besides Margin which do not support multi-threading can still be engineered to run in parallel whenever the genome has multiple chromosomes. Irrespective of the algorithm, the phasing operation can happen independently across chromosomes. Please check

<https://github.com/pjedge/longshot/issues/88>

<https://github.com/vibansal/longshot2#usage>

Unfortunately, due to the above reasons, I am fully convinced with the conclusions made in the paper.

Reviewer #3 (Remarks to the Author):

[Redacted]. I have no more major comments.

Minor comments:

If the vast majority of the running time is decompressing / parsing the BAM file, it should be possible to make KSNP even faster by parallelizing the work over the chromosomes.

Table 1: HapCUT2 running time with the HG001 CLR 50x dataset is likely wrong.

Software:

I managed to compile the software by following the instructions on Ubuntu / Intel / GCC. There were some compiler warnings that could be addressed. The software appeared to work correctly, but it printed the following message four times every time I ran it:

```
../build/ksnp: /lib/x86_64-linux-gnu/libhts.so.3: no version information available (required by
../build/ksnp)
```

On macOS / ARM / Clang with htslib installed from Homebrew, the first compilation attempt failed. The

second attempt was successful, once I realized I had to specify the htlib library directory instead of the installation directory (-DHTSLIB=/opt/homebrew/lib instead of -DHTSLIB=/opt/homebrew). There were again some compiler warnings that could be addressed. The software appeared to work correctly and without any issues.

Reviewer #5 (Remarks to the Author):

[Redacted].

To make the tool better accessible to the user, the authors may want to release it in dockerhub or suitable platforms.

The recent review article <https://genomebiology.biomedcentral.com/articles/10.1186/s13059-021-02328-9> might as well be helpful to readers.

Responses to reviewer comments

Reviewer #1	1
Reviewer #3	6
Reviewer #5	10

REVIEWER COMMENTS

Reply:Reviewer #1 (Remarks to the Author):

[Redacted]. I appreciate the inclusion of PacBio HiFi reads in the benchmark, as well as Supplementary Figure 1. I continue to believe that the proposed approach is novel with respect to existing methods for phasing. However, I am still not convinced if KSNP offers sufficient empirical advantage over the existing state-of-the-art.

Authors have presented the following argument in the Introduction section to motivate this work:

"As the amount of data increases, users typically use more threads or develop hardware acceleration solutions to improve the speed of read alignment and variant detection. However, The current phasing tools are still unable to fully utilize the parallel computing power of hardware, making it difficult to shorten the analysis time and gradually becoming an inefficient step in the pipeline"

The main results (Table 1 and Table 2) presented in the manuscript are based on sequential single-threaded runs of the existing phasing tools. However, the authors appear to ignore the fact that Margin supports efficient multi-threading. As a result, the performance evaluation has been done by using only a single thread. This evaluation is unfair because Margin would have used significantly less time with multiple threads. Please check:

<https://github.com/UCSC-nanopore-cgl/margin#resource-requirements>

Additionally, other tools besides Margin which do not support multi-threading can still be engineered to run in parallel whenever the genome has multiple chromosomes. Irrespective of the algorithm, the phasing operation can happen independently across chromosomes. Please check

<https://github.com/pjedge/longshot/issues/88>

<https://github.com/vibansal/longshot2#usage>

Unfortunately, due to the above reasons, I am fully convinced with the conclusions made in the paper.

Reply: Thank you very much for acknowledging the revisions made to the manuscript and recognizing the novelty of the KSNP algorithm.

Regarding the sentences cited by the reviewer, we apologize for the improper wording and appreciate the reviewer for pointing it out. In the revised version, we have modified the content to the following: “As the amount of data increases, users typically use more threads or develop hardware acceleration solutions to shorten the analysis time. More efficient algorithms are also desirable to reduce the computational costs associated with the pipeline” (Line135 – Line138).

We did not ignore the multi-threading capability of Margin and the potential to implement multi-processing for single-threaded phasing tools by splitting chromosomes. In fact, we fully utilized these time-saving approaches in our experiments. We apologize for not detailing this information in the original manuscript, which may have caused some misunderstandings. We would like to provide the following supplementary information in the revised manuscript:

We split the whole-genome BAM and VCF files by chromosomes. For Longshot, WhatsHap, HapCUT2, and KSNP, each chromosome was individually phased using a single thread. In the case of Margin, eight threads were used per chromosome for phasing tasks. The CPU time of all phasing tasks was then aggregated as the CPU time for each tool in the whole-genome phasing. We have revised the 'Methods' section of the manuscript to provide more descriptions on experimental methodology (Line354 - Line360 in Methods).

We used 'Running time' in the main text and tables, which is not precise. In the revised version, we have changed it to 'CPU time.' Additionally, we have included

Supplementary Tables 10 and 11, providing the CPU time and Wall clock time for the tools used in the experiments. CPU time is the amount of time a program is occupying and working on all CPU cores. It is the most commonly used metric for measuring CPU resource consumption. Wall clock time refers to the real-world time that the program completes its job and exits, including any delays and intermedia I/O time, without considering if multiple CPU cores are used. Considering that we have manually split the input BAM and VCF files by chromosomes and submitted the phasing tasks of each chromosome simultaneously to accelerate experimental processes, the wall clock time would be the maximum time taken for all tasks to complete, usually the wall clock time taken for chromosome 1 in human datasets. As shown in supplementary Tables 10 and 11, **when performing parallel phasing on split chromosomes and evaluating speed based on wall clock time, KSNP using 22 cores maintains a speed advantage of 3-15X compared to Longshot/Whatsap/HapCUT2, which also employs 22 cores. It retains a speed advantage of 9x compared to Margin, which utilizes 176 cores.** This fully validates the efficiency of KSNP.

It's important to note that wall clock time can be influenced by various factors, such as the system environment, availability of computational resources, and the method of submitting the computational tasks. In terms of fairness in evaluating the phasing tools, we believe that regardless of the multi-threading or multi-processing parallelism utilized by the tools, 'CPU time' is the most appropriate metric to reflect the time consumption. It represents the time actively spent by all threads on their respective cores. Although wall clock time represents the waiting time experienced by the user, it does not reflect the actual resource consumption or computational billing. While using multi-threading or data partitioning for multi-process parallelism reduces wall clock time or user waiting time, it does not result in true savings of computational resources. This is fundamentally different from the acceleration achieved by KSNP, which reduces computational complexity through algorithm optimization. Therefore, considering the computational resource consumption and computational billing, we believe that the evaluation of software speed should be based on CPU time rather than wall clock time.

We also strongly support the application of parallel techniques to reduce user waiting time. In fact, the '-c' parameter in KSNP allows users to specify individual chromosomes for phasing, enabling chromosome-level parallel processing without the need for splitting input files. We have also included this information in the software documentation to assist readers in its usage.

Supplementary Tables 10 CPU time and wall clock time for experiments in Table 1

Dataset	Tool	CPU core required ^a	CPU time ^b (s)	Wall clock time ^c (s)
HG001 CLR 50×	Longshot	22	13,485	1,182
	WhatsHap	22	20,714	1,850
	HapCUT2	22	6,100	537
	Margin	22 x 8	146,551	1,614
	KSNP	22	1,881	173
HG002 CLR 50×	Longshot	22	13,537	1,148
	WhatsHap	22	19,141	1,634
	HapCUT2	22	6,042	506
	Margin	22 x 8	128,479	1,404
	KSNP	22	1,607	152
HG005 CLR 50×	Longshot	22	17,968	1,565
	WhatsHap	22	22,968	2,036
	HapCUT2	22	8,202	722
	Margin	22 x 8	145,455	1,668
	KSNP	22	2,088	194
HG01109 CLR 40×	Longshot	22	20,794	1,728
	WhatsHap	22	22,919	1,915
	HapCUT2	22	9,615	810
	Margin	22 x 8	125,506	1,362
	KSNP	22	1,304	124

Supplementary Tables 11 CPU time and wall clock time for experiments in Table 2

Dataset	Tool	CPU core required ^a	CPU time ^b (s)	Wall clock time ^c (s)
HG001 ONT 50×	Longshot	22	21,249	1,890
	WhatsHap	22	19,989	1,748
	HapCUT2	22	10,284	953
	Margin	22 x 8	151,309	1,703
	KSNP ^a	22	2,091	215
HG002 ONT 50×	Longshot	22	30,954	2,644
	WhatsHap	22	23,909	1,968
	HapCUT2	22	15,645	1,304
	Margin	22 x 8	169,673	1,847
	KSNP	22	2,265	208
HG005 ONT 50×	Longshot	22	32,509	3,464
	WhatsHap	22	22,990	2,056
	HapCUT2	22	16,633	1,924
	Margin	22 x 8	192,759	2,092
	KSNP	22	2,558	256

a. The input whole-genome BAM and VCF files of human dataset were split into 22 chromosomes. For Longshot, WhatsHap, HapCUT2, and KSNP, each chromosome was individually phased using a single thread. For Margin, eight threads were used per chromosome for phasing tasks.

b. CPU time is the sum of CPU time for each chromosome phasing task.

c. Wall clock time is the real time of the longest-running chromosome phasing task, usually the wall clock time of chromosome 1.

Reviewer #3 (Remarks to the Author):

[Redacted]. I have no more major comments.

Reply: Thank you very much for reviewing our manuscript again. We appreciate your suggestions for both the manuscript and the software.

Minor comments:

If the vast majority of the running time is decompressing / parsing the BAM file, it should be possible to make KSNP even faster by parallelizing the work over the chromosomes.

Reply: Thanks for the suggestion. In fact, we have utilized this time-saving approaches in our experiments. We have manually split the input BAM and VCF files by chromosomes and submitted the phasing tasks of each chromosome simultaneously to accelerate experiments of the phasing tools. We have added the details in revised the 'Methods' section of the manuscript (Line354 - Line360 in Methods).

Additionally, the "-c" parameter in KSNP supports the analysis of specified chromosomes, making it convenient for readers to perform parallel analyses on multiple chromosomes. For KSNP, even when analyzing a single chromosome, the time spent on decompressing input files constitutes most of the overall analysis time. We have added the time cost of each chromosome in Supplementary Table 6.

Supplementary Table 6 Time consumption of each step of KSNP

Dataset	Read BAM&VCF (s)	Re- alignment (s)	DBG (s)	Output (s)	Total(s)) ^a
HG001 CLR 50x	1,576.9	166.7	68.6	9.8	1,879.6
HG002 CLR 50x	1,357.3	150.7	53.9	9.8	1,605.7
HG005 CLR 50x	1,753.6	213.9	72.5	9.3	2,086.3
HG01109 CLR 40x	1,019.5	151.9	105.8	12.4	1,302.8
HG001 ONT 50x	1,806.0	113.7	140.8	7.3	2,089.5
HG002 ONT 50x	1,890.3	133.2	217.2	7.3	2,264.0
HG005 ONT 50x	2,166.6	148.4	220.0	6.7	2,556.6
chr1 ^a	131.13	13.22	6.38	0.71	156.37
chr2	142.06	14.49	5.20	0.76	167.80
chr3	115.74	12.10	4.07	0.67	136.78
chr4	112.83	11.89	4.18	0.67	133.66
chr5	104.81	11.29	4.01	0.67	124.52
chr6	98.98	10.74	3.74	0.62	117.59
chr7	91.47	9.65	3.47	0.59	108.52
chr8	84.70	9.07	3.43	0.57	100.87
chr9	67.41	7.31	3.00	0.44	80.58
chr10	80.58	8.47	3.84	0.53	96.47
chr11	76.14	8.11	3.06	0.47	90.55
chr12	76.41	8.11	2.89	0.48	90.67
chr13	57.02	6.15	2.04	0.37	67.59
chr14	51.61	5.47	2.42	0.33	61.73
chr15	46.21	4.69	2.11	0.28	55.04
chr16	46.94	5.05	2.43	0.31	56.47
chr17	43.08	4.04	1.59	0.24	50.65
chr18	44.83	4.90	1.76	0.29	53.35
chr19	30.18	3.37	1.58	0.24	36.46
chr20	33.98	3.65	1.53	0.21	40.61
chr21	22.37	2.71	4.23	0.16	30.26
chr22	18.43	2.25	1.60	0.15	23.09

^aTime consumption of each chromosome is based on HG001-CLR-50x dataset.

Table 1: HapCUT2 running time with the HG001 CLR 50x dataset is likely wrong.

Reply: Thanks for pointing this out. We have corrected it in the revised manuscript.

Software:

I managed to compile the software by following the instructions on Ubuntu / Intel / GCC. There were some compiler warnings that could be addressed. The software appeared to work correctly, but it printed the following message four times every time I ran it:

```
../build/ksnp: /lib/x86_64-linux-gnu/libhts.so.3: no version information available  
(required by ../build/ksnp)
```

On macOS / ARM / Clang with htslib installed from Homebrew, the first compilation attempt failed. The second attempt was successful, once I realized I had to specify the htslib library directory instead of the installation directory (-DHTSLIB=/opt/homebrew/lib instead of -DHTSLIB=/opt/homebrew). There were again some compiler warnings that could be addressed. The software appeared to work correctly and without any issues.

Reply: We appreciate that you installed and tested KSNP.

The error message on Ubuntu/Intel/GCC might be caused by the wrong version of HTSLIB, which is required to be version v1.15.1 as declared in README.md file. We recommend the version of HTSLIB should be not less than v1.10.

Regarding the compilation on macOS/ARM/Clang, -DHTSLIB should specify the subdirectory containing libhts.so or libhts.so.3. Sorry for the trouble, and we are glad you had resolved it. We have included the install instruction in README.md file, as follows:

“Before KSNP installation, keep sure the dependency htslib has been correctly installed. Please add the subdirectory containing libhts.so to the default environmental variable for searching dynamic-link libraries (such as LD_LIBRARY_PATH

environment variable in Linux OS) or use the pre-compile option `-DHTSLIB=` when running `cmake` to specify the location.”

As to the compiler warnings, we have modified some source codes based on the information provided by the higher versions of compilers. We believe it would address your issues. We do anticipate more feedback from users on Github and are happy to help to solve their issues on various hardware platforms and operation systems.

We have recently released KSNP in Anaconda community where you can install it by `conda install` command. The relevant instructions have also been added to the README file on GitHub (<https://github.com/zhouqiansolab/KSNP>). Currently, KSNP is placed in our own Conda channel for the convenience of reviewers to test. After the article is published, KSNP can be officially pushed to the Bioconda channel.

Reviewer #5 (Remarks to the Author):

[Redacted].

To make the tool better accessible to the user, the authors may want to release it in dockerhub or suitable platforms.

Reply: Thanks for the suggestion! We have released KSNP in Anaconda community where genomic tools can be accessed and installed by a simple *conda install* command. Currently, KSNP is placed in our own Conda channel for the convenience of reviewers to test. After the article is published, KSNP can be officially pushed to the Bioconda channel. The relevant instructions have also been added to the README file on GitHub (<https://github.com/zhouqiansolab/KSNP>).

The recent review article <https://genomebiology.biomedcentral.com/articles/10.1186/s13059-021-02328-9> might as well be helpful to readers.

Reply: Thanks for the suggestion. We have included this article as a reference in our revised manuscript (Line142 in main text and ref 15 in References).

Reviewer #1 (Remarks to the Author):

The main contribution of the manuscript (as highlighted in the Abstract) is "5-81x speedup" compared to other tools. Taking this into account, it becomes extremely important that the measurement of runtimes of different tools is done precisely.

I thank reviewers for taking time and giving detailed response to my comments. However, the response still does not address my concerns. I still feel that experiments are inconsistent with the conclusions of the paper. As a practitioner, if I read a paper that describes a new method which is say 5x faster than existing methods, I immediately assume that waiting time experienced by the user is going to reduce by 5x. If I am using cloud resources (e.g., AWS), I would also expect an equivalent reduction in the cost. However, none of this is true here.

I'm sorry but I have to disagree with the following points mentioned in the authors' response:

- "We believe that the evaluation of software speed should be based on CPU time rather than wall clock time."
- "CPU time is the most appropriate metric to reflect the time consumption"
- "Although wall clock time represents the waiting time experienced by the user, it does not reflect the actual resource consumption or computational billing."

The correct way to benchmark the runtime of tools is to measure the end-to-end elapsed time (i.e., wall-clock time) of each tool. This would require getting **exclusive** access to a multicore computer server, and running each method one by one on that server. Every method that supports parallel computing should be allowed to use multiple threads. One way to measure end-to-end time on Linux machines is to use `/usr/bin/time` command which reports "elapsed time" in the end. If a single tool requires execution of multiple commands, then the total elapsed time can be added for each task. This time would automatically include both computing time as well as input/output (IO) time. This quantity should eventually reflect the total waiting experienced by the user if they run the tool on an idle computer. In cloud (say AWS) also, for a given instance type, the total wall-clock time multiplied with the AWS instance price would give you the total amount charged to the user.

For more details about the correct benchmarking methodology and measurement of the cloud cost, please see:

Shafin, K., Pesout, T., Lorig-Roach, R. et al. Nanopore sequencing and the Shasta toolkit enable efficient de novo assembly of eleven human genomes. *Nat Biotechnol* 38, 1044–1053 (2020).
<https://doi.org/10.1038/s41587-020-0503-6>

Unfortunately, the benchmarking methodology used in the manuscript is incorrect:

- (a) The main manuscript quantifies speedup by looking at CPU times. This should have been done on the basis of end-to-end wall clock time. Tables 1 and 2 in the main text should also report the wall-clock time instead of CPU times.
- (b) Even though the supplementary tables 10 and 11 are now added to report wall-clock time, the reported time appears incorrect to me. For example, how can KSNP complete processing of a large HG001 ONT 50x dataset in 215 seconds (Supp. Table 11) when the "read-in and decompression time of the input is about ~30 minutes (main text, line 116). Please note that input/output time cannot be ignored when reporting the total application time.
- (c) I would agree that both CPU times and wall-clock time are useful to know. However, authors should either communicate both parameters in the main text, or for brevity, they can just report the end-to-time wall-clock time. Note that CPU time excludes IO time.
- (d) In Supplementary Tables 10 and 11, please use "Threads used" instead of "CPU core required". Cores and threads are different.

In my view, the overall benefit of the proposed method may diminish after the speedup is recalculated

properly. All my observations are based on reading the authors' response and manuscript. I have not tested the software.

Reviewer #1 (Remarks to the Author):

The main contribution of the manuscript (as highlighted in the Abstract) is "5-81x speedup" compared to other tools. Taking this into account, it becomes extremely important that the measurement of runtimes of different tools is done precisely.

I thank reviewers for taking time and giving detailed response to my comments. However, the response still does not address my concerns. I still feel that experiments are inconsistent with the conclusions of the paper. As a practitioner, if I read a paper that describes a new method which is say 5x faster than existing methods, I immediately assume that waiting time experienced by the user is going to reduce by 5x. If I am using cloud resources (e.g., AWS), I would also expect an equivalent reduction in the cost. However, none of this is true here.

I'm sorry but I have to disagree with the following points mentioned in the authors' response:

- "We believe that the evaluation of software speed should be based on CPU time rather than wall clock time."
- "CPU time is the most appropriate metric to reflect the time consumption"
- "Although wall clock time represents the waiting time experienced by the user, it does not reflect the actual resource consumption or computational billing."

Reply: We apologize for the potentially inaccurate description of the roles of CPU time and wall clock time in our previous response. Based on the suggestions from the reviewers and editors, in this revision, we have included both CPU time and wall clock time in the main text and the supplementary tables to facilitate readers' evaluation of computational resource consumption (Line 116 - 121).

The correct way to benchmark the runtime of tools is to measure the end-to-end elapsed time (i.e., wall-clock time) of each tool. This would require getting *exclusive* access to a multicore computer server, and running each method one by one on that server. Every method that supports parallel computing should be allowed to use multiple threads. One way to measure end-to-end time on Linux machines is to use `/usr/bin/time` command which reports "elapsed time" in the end. If a single tool

requires execution of multiple commands, then the total elapsed time can be added for each task. This time would automatically include both computing time as well as input/output (IO) time. This quantity should eventually reflect the total waiting experienced by the user if they run the tool on an idle computer. In cloud (say AWS) also, for a given instance type, the total wall-clock time multiplied with the AWS instance price would give you the total amount charged to the user.

For more details about the correct benchmarking methodology and measurement of the cloud cost, please see: Shafin, K., Pesout, T., Lorig-Roach, R. et al. Nanopore sequencing and the Shasta toolkit enable efficient de novo assembly of eleven human genomes. *Nat Biotechnol* 38, 1044–1053 (2020). <https://doi.org/10.1038/s41587-020-0503-6>

Reply: We appreciate the reviewers' suggestions regarding the calculation method for wall clock time.

The experiments in this study were conducted on a high-performance computing cluster node with 24 cores. To shorten the analysis time, we split the input BAM and VCF files of human genome into 22 parts by chromosomes and submitted 22 separate computational tasks. For single-threaded tools such as WhatsHap, HapCUT2, Longshot, and KSNP, all the 22 tasks were submitted simultaneously. As for margin, we used 8 threads within each task, and the 22 tasks were executed sequentially. The CPU time and end-to-end wall clock time for each task were recorded using the '/usr/bin/time' command.

Following the method and the reference paper suggested by the reviewer, we recalculated the wall clock time by summing up the wall clock times of the 22 tasks. For single-threaded tools like WhatsHap, HapCUT2, Longshot, and KSNP, the wall clock time slightly exceeds the CPU time as pointed out by the reviewer that CPU time excludes IO time. Compared to WhatsHap, HapCUT2 and Longshot, KSNP demonstrates a speed advantage of 5-11x in terms of both CPU time and wall clock time on seven human datasets. For Margin, which utilizes 8 threads, the wall clock time is approximately 1/8 of the CPU time, which is reasonable. KSNP demonstrates 81x and 10x speedups in terms of CPU time and wall clock time, respectively.

Based on the latest results, we have updated the main text and tables in the revised manuscript.

Unfortunately, the benchmarking methodology used in the manuscript is incorrect:

(a) The main manuscript quantifies speedup by looking at CPU times. This should have been done on the basis of end-to-end wall clock time. Tables 1 and 2 in the main text should also report the wall-clock time instead of CPU times.

Reply: Many thanks for the suggestion. We have included both CPU time and wall clock time in the revised tables. Additionally, when describing the speedup in the main text, we have instead used wall clock time.

(b) Even though the supplementary tables 10 and 11 are now added to report wall-clock time, the reported time appears incorrect to me. For example, how can KSNP complete processing of a large HG001 ONT 50x dataset in 215 seconds (Supp. Table 11) when the "read-in and decompression time of the input is about ~30 minutes (main text, line 116). Please note that input/output time cannot be ignored when reporting the total application time.

Reply: Apologies for the previous unclear presentation of the results. The '215 seconds' was the wall clock time of the phasing task for HG001 chr1, while the '30 minutes' represented the read-in and decompression time in the phasing of whole HG001 genome.

In this revised manuscript, we have recalculated the wall clock time for all of the experiments following the method suggested by the reviewer. In the phasing of the HG001 ONT 50x dataset, the end-to-end wall clock time of KSNP is 2,307 seconds (~38 min, Table 1) and the read-in and decompression time of the input files is 1,993 seconds (~33 min, Table S6).

(c) I would agree that both CPU times and wall-clock time are useful to know. However, authors should either communicate both parameters in the main text, or for brevity, they can just report the end-to-time wall-clock time. Note that CPU time excludes IO time.

Reply: In this revised manuscript, we have included both CPU time and wall clock time to facilitate the evaluation of the computational resource consumption.

(d) In Supplementary Tables 10 and 11, please use "Threads used" instead of "CPU core required". Cores and threads are different.

Reply: In this revised manuscript, the wall clock time is presented in Tables 1 and 2. Accordingly, the Supplementary Tables 10 and 11 have been removed. We have carefully revised the descriptions of CPU cores in the main text to 'threads' instead.

In my view, the overall benefit of the proposed method may diminish after the speedup is recalculated properly. All my observations are based on reading the authors' response and manuscript. I have not tested the software.

Reply: Based on the latest calculations of CPU time and wall clock time, KSNP achieves a speedup ratio of 5-11x compared to WhatsHap, HapCUT2 and Longshot, all of which are single-threaded tools. Against Margin, which utilizes 8 threads in our experiments, the speedup ratio of KSNP is ~10x. KSNP utilizes fewer cores and less computation time while achieving comparable quality of computational results. KSNP enables users to perform genome phasing using lower-end, single-core machines, resulting in significant cost savings for users.

The speed advantage of KSNP is attributed to its innovative approach to haplotype assembly, which should not be overlooked. KSNP employs the de Bruijn graph (DBG) widely used in *de novo* genome assembly for SNP haplotype assembly, fully leveraging the ability of DBG in handling high-throughput and noisy reads. This differs significantly from the existing phasing methods. The success of KSNP demonstrates the potential of DBG in handling long reads, which can be inspiring for the development of analysis methods for long reads.

Reviewer #6 (Remarks to the Author):

I am fine with regard to how the authors have addressed the points raised by Reviewer 1.